# Cost-analysis of real time RT-PCR test performed for COVID-19 diagnosis at India's national reference laboratory during the early stages of pandemic mitigation

**Naveen Minhas** [1], **Yogesh K. Gurav** [1]*, **Susmit Sambhare** [1], **Varsha Potdar**[2], **Manohar Lal Choudhary**[2], **Sumit Dutt Bhardwaj**[2], **Priya Abraham**[3]

1 Health Technology Assessment Resource Centre (HTA-RC), Dengue & Chikungunya Group, ICMR-National Institute of Virology, Pune, Maharashtra, India, 2 Human Influenza Group, National Influenza Centre, ICMR-National Institute of Virology, Pune, Maharashtra, India, 3 ICMR-National Institute of Virology, Pune, Maharashtra, India

☯ These authors contributed equally to this work.

* gurav.yk@gmail.com

**Data Availability Statement:** All relevant data are within the paper and its Supporting Information files.

## Abstract

Real-time reverse transcription polymerase chain reaction (rRT-PCR) is one of the most accurate and extensively used laboratory procedures for diagnosing COVID-19. This molecular test has high diagnostic accuracy (sensitivity and specificity) and is considered as the gold standard for COVID-19 diagnosis. During COVID-19 surge in India, rRT-PCR service was encouraged and supported by the government of India through existing healthcare setup at various levels of healthcare facilities. The primary purpose of this research was to determine the per-unit cost of providing COVID-19 rRT-PCR services at the national reference laboratory at ICMR-National Institute of Virology in Pune during the early phase of COVID-19 pandemic mitigation, from the provider's perspective. The monthly cost for rRT-PCR testing as well as an estimated annual average unit cost for testing that takes account of peaks and troughs in pandemic were investigated. The time frame used to estimate unit cost was one year (July 2020-June 2021). For data collection on all resources spent during the early phase of pandemic, a conventional activity-based bottom-up costing technique was used. Capital costs were discounted and annualized over the estimated life of the item. Apportioning statistics were selected for cost heads like human resources, capital, and equipment based on time allocation, sharing of services, and utilization data. The data was also used to understand the breakdown of costs across inputs and over time and different levels of testing activity. During the initial phase of pandemic mitigation, the per unit cost of providing the COVID-19 rRT-PCR test was estimated to be ₹566 ($7.5) in the month of July 2020, where the total 56318 COVID-19 rRT-PCR tests was performed. The major proportion (87%) of funds was utilized for procuring laboratory consumables, followed by HR (10%), and it was least for stationary & allied items (0.02%). Unit cost was found to be the most sensitive to price variations in lab consumables (21.7%), followed by the number of samples tested (3.9%), salaries paid to HR (2.6%), price of equipment (0.23%), and building rental price (0.14%) in a univariate sensitivity analysis. The unit cost varies over the period

**Funding:** The author(s) received no specific funding for this work.

**Competing interests:** The authors have declared that no competing interests exist.

of the pandemic in proportion with the prices of consumables and inversely proportional with number of tests performed. Our study would help the Government to understand the value for money they invested for laboratory diagnosis of COVID-19, budget allocation, integration and decentralization of laboratory services so as to help for achieving universal health coverage.

## Introduction

Corona viruses are emerging as a threat to people in the 21st century. COVID-19 is the second pandemic the world is facing in the 21st century after the H1N1 influenza pandemic in the year 2009 and demonstrates how rapidly a new virus can spread to every part of the globe. In the first year of this pandemic, the world statistics showed 64 million people have been affected by this malaise, and the global economy has experienced a loss of more than $1 trillion [1]. Early diagnosis of suspect case of COVID-19 is very crucial and plays a pivotal role to contain the spread of this disease in community [2, 3]. This disease can be diagnosed by various imaging (chest X-ray, CT scan, Pulmonary ultrasonography) tests. However, real time reverse transcriptase polymerase chain reaction (rRT-PCR) based on molecular assays regarded as a gold standard for its laboratory diagnosis by World Health Organization (WHO) [4–6]. High sensitivity was observed by rRT-PCR test as compared to other diagnostic tests for COVID-19 [7]. However, for performing a rRT-PCR test, necessary kits & reagents with compatible RT-PCR machines, well equipped laboratory is required in addition to skilled manpower.

In the prevailing COVID-19 situation, Indian government has expend most of health budget towards the provisioning of rRT-PCR test like in terms of building up COVID-19 rRT-PCR labs, recruiting manpower, procurement of instruments and shipping of chemicals & reagents. India has spent over Rs 100 crore on COVID-19 testing in both government and private labs and takes into account only rRT-PCR tests, which confirm the COVID-19 infection, as reported by Times of India [8]. The Economic Survey by Ministry of Finance, Govt. of India, mentioned that the health sector was the worst hit by this pandemic and Expenditure on health sector increased from Rs. 2.73 lakh crore in 2019–20 (pre-COVID -19) to Rs. 4.72 lakh crore in 2021–22, an increase of nearly 73% [9]. The expenses made on rRT-PCR testing have major impact on health budget allocated for this pandemic management. Therefore, accurate cost data is essential parameter for economic and financial evaluations which further help decision makers to take wise decision for efficient resource allocation. The financial evaluation executed to assess the adequate incremental cash flows to recover the financial costs without external support while the economic analysis is carried out with societal perspective and reflect the true value of the project to society. In economic evaluations, all positive and negative benefits are included and quantified in monetary terms [10].

A precise and reliable costing data can be generated from micro-costing method which involves "direct enumeration and costing out of every input consumed in providing rRT-PCR service" [11–13]. The cost of rRT-PCR test for COVID-19 has been depicted in social media and newspapers since early 2020; but, these are the prices with inclusion of profit. Such price data may have mostly depend on the business model of provider and does not reflect actual production cost of lab test. The actual production cost of test is primary and essential requirement of economic evaluation study and also beneficial for better resource allocation. In Micro-costing approach cost related to each and every resources consumed is employed to estimate unit cost; thus, micro-costing reflects true cost to society and healthcare system.

The government of India is planning to functionalise four regional virology laboratories across the country [14]. There is necessity for understanding the cost of the laboratory tests for diagnosis the viral etiology at research institutes by keeping in the mind that most of the time many clinical samples used to be referred by the local state government to reference virology institutes so as to providing quick diagnosis. The cost data is primary necessity for a judgement of adequate investment and resource allocation at the time of planning new research laboratories [12]. Despite the fact that India has a number of Virus Research Diagnostic Laboratory (VRDL) network, unit test costs are not readily available. By using a micro-costing approach, we estimated the per unit cost of the COVID-19 rRT-PCR test performed for laboratory diagnosis of COVID-19 at the Indian Council of Medical Research—National Institute of Virology (ICMR-NIV), Pune (Maharashtra State), India, which is a reference laboratory for doing virology research in India.The cost of conducting a single COVID-19 rRT-PCR test is referred to as the unit cost. The estimated cost per test is particularly relevant to pandemic mitigation efforts in the early stages.

## Methodology

### Study design and study site

From a provider's perspective, we estimated the per unit cost of COVID-19 rRT-PCR test using a bottom up micro-costing approach, considering fixed and variable costs. In Bottom up micro-costing each smallest component of resource used is estimated and aggregated for calculating unit cost [15].

The methodology followed attempts to measure per unit cost of the rRT-PCR test as accurately as possible by including all fixed and variable costs. Study site was the National Influenza Center (NIC), located at ICMR-NIV, Pune, as a major cost centre for data collection while administration & maintenance unit (AMU) was also included in this study as supportive cost centre. Costs were included starting from sample receipt at laboratory followed by sample sorting, sample separation, RNA extraction, rRT-PCR testing and reporting. The turnaround time for one rRT-PCR test was 3–4 hours. On an average nearly 19800 tests per month were carried out from July 2020 to June 2021 in NIC.

### Data collection

Data was collected in the proforma by a trained project staff after obtaining the necessary permissions. Cost data on both capital and recurrent resources was collected for the early pandemic period of July 2020. Data was collected from the cost center, which include the core diagnostic facility (NIC) [for laboratory diagnosis of COVID-19 at ICMR-NIV, Pune] and other supportive cost center [administration & maintenance unit (AMU)]. Capital resources include building space, laboratory equipments (rRT-PCR machines, biosafety cabinets, automated systems), other instruments (computer systems, furniture, and allied items) and all other resources that last for a period of more than one year. The cost of the building area was estimated by referring to the market rental price of locality. Recurrent resources are comprised of human resources (HR), laboratory consumables (rRT-PCR reagents and kits, plasticware and glass wares), non-laboratory consumables (stationary materials) and overhead expenses (utility bills). The data was collected from the stock registers, instrument log-books, attendance registers, duty rosters. The observational method was also used to collect information wherever documentary records were lacking. Comprehensive information about HR (numbers of staff, categories of staff and duty timings) was obtained from daily attendance registers & duty rosters. The gross monthly salary of HR was recorded from the pay bill register from the administration section of ICMR-NIV, Pune. All the staff members (regular, contractual,

project and outsourced daily wage staff) involved in laboratory tests were personally inter-viewed so as to get their time allocation data for the COVID-19 rRT-PCR test in proportion to the total working hours per day for different activities. All the respondents were interviewed after obtaining written informed consent. Stock registers were used to record the quantity of various consumables consumed during the reference period. Data on unit prices of consum-ables was obtained from indent registers, recent payment bills and confirmed with the pur-chase & bills sections of the institute. For selected consumable items (like plasticware, glassware, chemical & reagent), rate contract lists of the institute were also explored to get unit prices. Cost data on capital items like laboratory equipments, furniture and other non-con-sumable items were obtained from purchase & procurement records. The details of cost cen-tres and different sources of cost data are presented in Table 1.

Monthly bills for the reference period were used to account for utility expenses like electric-ity, water, telephone, bio-waste management, internet and laundry services. Data on utility costs was available for the entire institute as a whole, rather than for NIC and AMU separately. Electricity expenses were calculated in consultation with the engineering support unit of the institute using a list of equipment operated electrically, their time usage data per day, power consumed and energy consumed per day. Laundry bills were not generated for the reference period as the use of all reusable and recyclable items was strictly prohibited. Building infra-structure details were obtained from the engineering support unit of the institute and were ascertained room-wise along with the purpose for which they were being used.

## Allocation of resources/ apportioning statistics

Apportioning statistics were chosen for all cost heads based on time allocation, service sharing, and usage data, for example, records of job cards maintained by the engineering support unit were used to extract data for time devoted by engineering staff to repairing equipment. The details of the apportioning statistics used are given in Table 2. Other activity statistics, like the

**Table 1. Details of cost centres, cost heads, cost parameters and source of cost data utilized for calculating unit cost of COVID-19 rRT-PCR test.**

| Cost centers | Cost heads | Description | Cost parameter | Data source at ICMR-NIV |
|---|---|---|---|---|
| NIC* | Human resource | Scientific, technical & other supporting lab staff | Gross monthly salary | Salary slips |
| | Non-lab consumables | Stationary & allied items | Quantity consumed, unit price of consumable | Stock registers |
| | Lab consumables | lab consumables (glassware, plasticware, chemicals, reagents & kits etc.) | Quantity consumed, unit price of consumable | Stock registers, Purchase records, Rate contract lists |
| | Other instruments | Furniture & allied items | Total number, price of single unit | Facility survey, Purchase records |
| | Laboratory equipments | lab instruments | Total number, price of single unit | Facility survey, Purchase records |
| | Utilities | Overhead expenses (Electricity, Water, Internet, Telephone, Laundry and Bio-waste disposal etc.) | Monthly utility bills | Office records |
| | Physical Infrastructure | Building space/Area | Rental price per sq.ft. per month | Facility survey, Office record |
| AMU* | HR | Admin. staff & Engineering support staff | Gross monthly salary | Salary slips |
| | Utilities | Overhead expenses (Electricity, Water, Internet, Telephone) | Monthly utility bills | Office records |
| | Physical Infrastructure | Building space/Area | Rental price per sq. ft. per month | Facility survey, Office record |

*Abbreviations: National Influenza Centre (NIC); Administration and maintenance unit (AMU)

**Table 2. Allocation statistics used for various capital and recurrent resources.**

| Cost Head | Apportioning statistics used | Allocation criteria based on personal observation * (if any) | source of the data |
|---|---|---|---|
| **Human resources** | | | |
| NIC[#] | Total time spent by the staff in COVID-19 rRT-PCR based diagnosis | - | Interview |
| AMU[#] | Monthly duty hours for COVID related activities | During analysis only 50% of the total duty hours were taken as actual time devoted for COVID-19 rRT-PCR testing | Interview |
| **Consumables** | | | |
| Lab consumables | Based on Lab record | - | Records |
| Non-lab consumables | Based on Lab record | - | Records |
| **Equipments** | | | |
| Laboratory equipments and other instruments | The proportion of time used for COVID-19 rRT-PCR testing | - | Observation |
| **Utilities** | | | |
| NIC[#] | Human resource deployed and their time proportion for COVID-19 rRT-PCR testing | During analysis, only 80% of the total utilities were actually utilized for COVID-19 rRT-PCR testing | Records |
| AMU[#] | Human resource deployed and their time proportion for COVID-19 rRT-PCR testing | During analysis, only 50% of the total utilities were actually utilized for COVID-19 rRT-PCR testing | Records |
| **Building space** | | | |
| NIC[#] | Human resource deployed and their time proportion for COVID-19 rRT-PCR testing | Only 80% of the building space were actually utilized for COVID-19rRT-PCR testing | Records |
| AMU[#] | Human resource deployed and their time proportion for COVID-19 rRT-PCR testing | Only 50% of the total utilities were actually utilized for COVID-19 rRT-PCR testing | Records |

* Personal observation by the investigators during data collection and routine facility survey.

[#]Abbreviations: National Influenza Centre (NIC); Administration and maintenance unit (AMU)

total number of COVID-19 rRT-PCR tests performed in a reference period, were taken from the laboratory test record registers at NIC. The majority of the NIC staff has devoted their full duty hours to the COVID-19 rRT-PCR test during the reference period. However, some MTS staff, like sweepers and staff deputed from other departments, whose service was jointly shared between different departments; were apportioned based on their time allocation. The HR costs incurred by AMU were allocated based on total number of staff deployed for the COVID-19 related work, the monthly duty hours devoted to the COVID-19 related work and daily observation [Table 2]. Each room area of both the cost centers was apportioned based on HR and time devoted per day for COVID-19 diagnosis in that particular room [Table 2]. The space cost which was jointly shared for the COVID-19 rRT-PCR test and other routine scientific activities was apportioned based on personal observation, i.e. the building space of NIC was actually utilized at 80% for the COVID-19 rRT-PCR test while the rest (20%) was for other routine scientific activities. In the case of AMU, the building space cost was apportioned based on the HR deployed for COVID-19 related work and their time allocation per day. The sources of data used for selecting allocation criteria in this study are described in Table 2. Cost of equipments were apportioned based on their actual time usage for COVID-19 rRT-PCR testing.

## Data analysis

All data collected from different cost centers was entered into Microsoft Office Excel for cost analysis. The unit cost of the COVID-19 rRT-PCR test was calculated using an activity-based bottom-up micro-costing approach [16]. Activity-based costing has high granularity as it

enlists, quantifies, and values every single item required for providing service. The currency was converted from Indian rupees (₹) to US dollars ($) as per the exchange rates of July 2020 [17] as the prices of most of the laboratory and other consumables included in this study were available for July 2020.

For those HR (laboratory staff) who worked jointly on different activities (e.g. staff involved in multidisciplinary work like COVID-19 rRT-PCR test, kit validation, shipment of COVID-19 diagnostic kits and routine diagnosis of influenza and other respiratory viruses),we estimated the time contribution by the staff solely for the COVID-19 rRT-PCR test. This relative time contribution was then multiplied with the gross salary of the staff member to elicit the cost of HR for the COVID-19 rRT-PCR test.

Capital cost was annualized by considering the average life span of the capital items to arrive at the equivalent annual cost. A discount rate of 3% was applied in accordance with the guidelines given by the International Society for Pharmacoeconomics and Outcome Research for India [18]. The useful life of buildings and structures was considered 20 years [19]; the useful life of other capital items was usually taken as 5 years and, in some instances, selected on the perception of lab staff about the same. We calculated building space costs by multiplying the estimates of floor area (sq.ft.) of rooms devoted to the COVID-19 rRT-PCR test with local commercial rental prices of similar space. For other capital resources like lab instruments, furniture and allied items, the original purchase price and year of purchase were traced from the record books maintained by the store section; otherwise, the fixed rates in recent government contracts for purchasing instruments and furniture were used. Missing costs of equipment and other goods were also obtained from local vendors and from relevant websites (Indiamart, GeM portal) on the internet [20, 21]. Costs incurred on recurrent resources (like rRT-PCR enzyme kits, primer-probe mixes, and all other lab & non-lab consumables) were estimated by multiplying the unit prices with the resources consumed in the reference period.

## Unit cost estimation

To calculate the unit cost of the COVID-19 rRT-PCR test, we used the average cost method, i.e. the total operating cost was divided by the number of COVID-19 samples tested in July 2020. The unit cost of providing the COVID-19 rRT-PCR test during the early stages of pandemic mitigation was calculated using the data collected for the month of July 2020. Subsequently, the data was used to derive the total cost for the period of July 2020 to June 2021 and for estimating the unit cost as well as for all further analysis based on the resulting annual data.

The costs were classified into fixed and variable costs, as some costs are constant and did not change with respect to different output levels and the latter one behave differently with respect to different output levels and also the nature of the pandemic has led to fluctuations in output. The fixed and variable costs were estimated using HR, equipments and building space as fixed cost components, while consumables were considered a variable component. There were some semi-variable costs, like utility (electricity), for which some components were fixed and others were variable based on consumption. Total cost for time period July 2020 to June 2021 for providing the COVID-19 rRT-PCR test was derived mathematically in which the fixed components were kept constant over the year while the variable components were varied in proportion to the number of COVID-19 samples tested in a year.

Further analysis was performed to reveal cost distribution between recurrent and capital cost. In this analysis, equipments and building space were considered as capital items while HR, consumables, and utilities were kept under recurrent items. The detailed analysis of costs breakdown under consumables, utilities, and equipments was performed to understand major contributors of costs. Regression analysis was done to determine the relationship between

number of samples tested and unit costs of COVID-19 rRT-PCR test for the period of July 2020 to June 2021.

## Sensitivity analysis

A univariate sensitivity analysis for unit cost was carried out using annual data wherein the base value of HR salaries, prices of laboratory consumables, prices of equipment, monthly rental price for building space and the number of the COVID-19 samples tested were varied by 25% on both sides. We also estimated the sensitivity of the per unit cost for providing the COVID-19 rRT-PCR test to variations in discount rates, i.e. at 3% and 10%. To incorporate the wide variance in pricing of laboratory consumables during the initial phases of country's COVID-19 outbreak and intervention like cost-capping on consumables by the government, we adjusted the pricing of laboratory consumables by up to 75% in the sensitivity analysis.

## Ethical statement

This study was approved by the ICMR-National Institute of Virology, Pune Institutional Ethics Committee (No: NIV/IEC/May/2020/D-4). The present cost analysis study does not involve any ethical concerns of human participants, samples or data from human subjects. The specimens collected at various hospital were sent to the study sites for the diagnosis purpose. The cost of the tests is being analyzed in this study. Hence as per the institute's ethics committee guidelines, there is no need to have a consent (written or verbal) from the study participant.

## Results

### Total cost and per unit cost

In the current study, the unit cost of providing the COVID-19 rRT-PCR test during the early stages of pandemic mitigation was estimated to be ₹566 ($7.5). The total number of COVID-19 rRT-PCR tests performed in the reference month was 56318. The total annual operating cost was estimated to be approximately ₹164.4 million ($2.2 million) and the unit cost ₹691 ($9.2) for the 237892 COVID-19 samples tested in a reference year. Majority of the funds (87%) were utilized for procuring lab consumables followed by HR (10%) [Table 3].

### Recurrent costs

The total number of HR deployed for the COVID-19rRT-PCR test was 101, which included71 from NIC [10 scientific staff, 35 technical staff, 11 multi tasking staff (MTS), 15 data entry staff]and 30 from AMU. Of which, 49 (48.5%) staff were on regular employment and the rest 52 (51.5%) were on contract/project mode/hired on daily wages. The total annual apportioned cost incurred by HR towards the COVID-19 rRT-PCR test was ₹16943111 ($226119.2) which includes the salary of NIC staff [₹16400932 ($218883.4)]and AMU staff [₹54279 ($724.4)]. The details of the associated HR and costs incurred by them are given in Table 4.

The total annual costs incurred by lab and non-lab consumables were approximately ₹143111722 ($1909938.9) and ₹27964 ($373.2) respectively. The majority of the funds were utilized for procuring lab consumables, which comprise rRT-PCR reagents & kits [₹127083209 ($1696025.7)] and plasticwares [₹16028512 ($213913.1)] (S1A Fig). Among rRT-PCR reagents and kits, the RNA extraction kit, MagMax COVID-19 viral RNA Isolation kit [₹76783675 ($1024738.8)], was the most expensive, followed by the rRT-PCR enzyme kit, Superscript III Platinum one step qRT-PCR kit [₹46220163 ($616844.6)] (S1B Fig).On the other hand, in the case of non-lab consumables, funds (0.01%) were mainly utilized towards the paper work done in maintaining office & lab records, data entry and report generation.

**Table 3. Total cost and unit cost distribution in Indian rupees (₹) and US dollar ($) among various cost heads for COVID-19 rRT-PCR test during July 2020 to June 2021.**

| Sr No. | | Cost Head | Total Cost (₹) | Total Cost ($) | Per Unit cost (₹) | Fund distribution (%) |
|---|---|---|---|---|---|---|
| 1 | | **HR** | 16,943,112 | 226,119 | 71.2 | 10.3 |
| 2 | | **Non-lab consumables** | 27,964 | 373 | 0.1 | 0.02 |
| 3 | | **Lab consumables** | 143,111,722 | 1,909,939 | 601.6 | 87.0 |
| | a | RT-PCR kits and reagents | 127,083,210 | 1,696,026 | 534.2 | 77.26 |
| | b | Plastic ware | 16,028,513 | 292,620 | 67.4 | 9.74 |
| 4 | | **Laboratory equipments** | 1,144,894 | 1,5279 | 4.8 | 0.7 |
| | a | rRT-PCR machine | 631,981 | 8,434 | 2.7 | 0.38 |
| | b | QiAgility system | 242,488 | 3,236 | 1.0 | 0.15 |
| | c | MagMax RNA extractor machine | 162,231 | 2,165 | 0.7 | 0.10 |
| | d | Other equipment's | 108,192 | 1,444 | 0.5 | 0.07 |
| 5 | | **Other instruments** | 411,241 | 5,488 | 1.7 | 0.25 |
| 6 | | **Utility expenses** | 1,904,466 | 25,417 | 8.0 | 1.16 |
| | a | Electricity | 1,482,748 | 19,788 | 6.2 | 0.90 |
| | b | Water | 336,399 | 4,490 | 1.4 | 0.20 |
| | c | Telephone | 81,909 | 1,093 | 0.3 | 0.05 |
| | d | Biowaste | 3,410 | 46 | 0.01 | 0.002 |
| 7 | | **Building space** | 954,078 | 12,733 | 4.0 | 0.57 |
| | | **Total (1+2+3+4+5+6+7)** | **164,497,877** | **2,195,354** | **691.5** | **100** |

Total cost incurred by utility costs (which included energy and water bills as well as internet, telephone, and bio-waste disposal) was ₹1904866 ($25421.9). Electricity accounted for the vast majority (77.8%) of utility costs, followed by bio-waste disposal (17.7%), water (4.3%), and telephone services (0.17%) (S1C Fig).

## Capital costs

Laboratory equipment and other instruments incurred yearly costs of ₹1144893 ($15279.5) and ₹411240 ($5488.3), respectively. 55.2% of the cost of laboratory equipment was incurred by rRT-PCR machines, followed by 21.2% and 14.2%, respectively, by QiAgility systems and MagMax RNA extractor machines (S1D Fig). Among other instruments, significant

**Table 4. Details of human resource and their cost contribution towards COVID-19 rRT-PCR test.**

| Cost Centre | Employment type | Number of staff | | | | | Cost contribution (₹) | Percentage distribution |
|---|---|---|---|---|---|---|---|---|
| | | Scientific staff | Technical staff | Multi tasking staff | Data entry staff | Admin. & maintenance staff | | |
| **NIC** | Regular | 5 | 20 | 1 | 0 | 0 | 10,584,362.0 | 62.5 |
| | Contract | 5 | 15 | 10 | 15 | 0 | 5,816,570.3 | 34.3 |
| **AMU** | Regular | 0 | 0 | 0 | 0 | 23 | 530,319.4 | 3.1 |
| | Contract | 0 | 0 | 0 | 0 | 7 | 11,860.8 | 0.07 |
| **Total** | | **10** | **35** | **11** | **15** | **30** | **16,943,111.9** | **100.00** |
| | Cost contribution (₹) | 3,804,125 | 10,366,711 | 953,884 | 1,276,864 | 541,529 | 16,943,112 | |
| | Percentage distribution | 22.5 | 61.2 | 5.6 | 7.5 | 3.2 | **100** | |

contributions were furniture and related things (42%), electronic items (35%), and electrical items (35%). Heavy-duty instruments (such as indoor and outdoor units, puff panels, and heaters) necessary to maintain walk-in refrigeration chambers and specialized rooms such as negative pressure rooms were placed under other instruments and accounted for around 20% of the overall cost incurred.

Building area of 2661.28 square feet (sq.ft) was used in the NIC for lab tests which include 12 rooms of different dimensions. In contrast, the total area covered by AMU was 2070 sq.ft comprises 5 rooms. Building space cost for the testing of COVID-19 samples was 0.5% of the total cost [₹954078 ($12732.9)] among which 96% was contributed by NIC [₹912194 ($12173.9)] and 4% by AMU [₹41884 ($559.0)].

## Fixed & variable costs

Almost 72% [₹118749735 ($1584809.0)] of the total annual operating cost, was the variable cost while the remaining 28% [₹45746497 ($610523.1)] was fixed cost. Human resources and lab consumables were the major cost components among fixed and variable costs, respectively (Fig 1).

## Sensitivity analysis

When the input values are changed by 25% on the lower and upper sides, the unit cost ranges from [₹541.1 ($7.2) to ₹841.9 ($11.2) for lab consumables, [₹719.0 ($9.6) to ₹675.0 ($9.0)] for the number of samples tested, [₹673.7 ($9.0) to ₹709.3 ($9.5)] for HR salaries, [₹689.8 ($9.2) to ₹693.1 ($9.2)] for equipment price, and [₹690.5 ($9.2) to ₹692.5 ($9.2) for rental price of building. The unit cost was found to be most sensitive to variation in the price of lab consumables (21.7%), followed by the number of samples tested (3.9%), salaries paid to HR (2.6%), price of equipment (0.23%) and the rental price for the building (0.14%). A tornado diagram showing the sensitivity of unit cost to different input parameters are given in Fig 2.

Among lab consumables prices, unit cost was mainly sensitive to variation in prices of COVID-19 rRT-PCR kits & reagents followed by rRT-PCR plastic wares and other lab consumables. The 25% variation in the COVID-19 rRT-PCR kits & reagents leads to 19.3% change

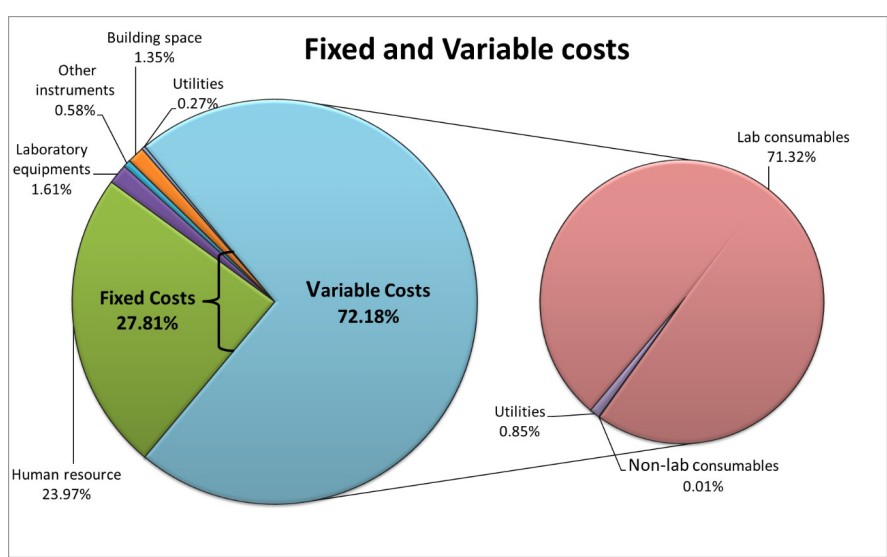

**Fig 1. Fixed & variable costs distribution for COVID-19 rRT-PCR test.**

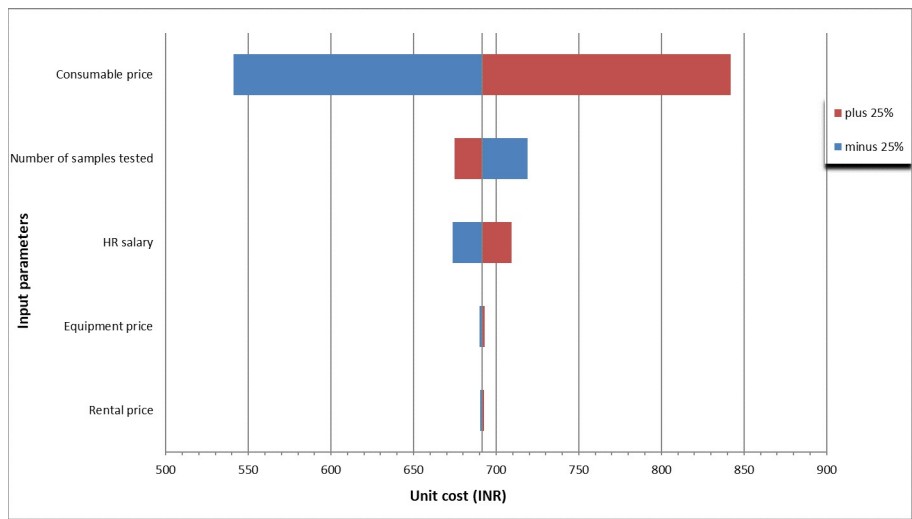

**Fig 2. Tornado diagram for sensitivity analysis of different input parameters for COVID-19 rRT-PCR test.**

in unit cost of RT-PCR. Similarly, 25% change in rRT-PCR plastic wares and other lab consumables results in 2.1% and 0.38% variation in unit cost respectively. While in case of salaries paid to HR, it was most sensitive to variation in salary structure of technical staff (leads to 1.8% change in unit cost) followed by scientific staff (leads to 0.57% change in unit cost).With the change in discount rates, the unit cost did not change much as compared to other parameter variation. The unit cost varies from ₹691.5 ($9.2) to ₹693.6 ($9.3) depending on the discount rate, which ranges from 3% to 10%. Fig 3 shows the regression equation used to establish a

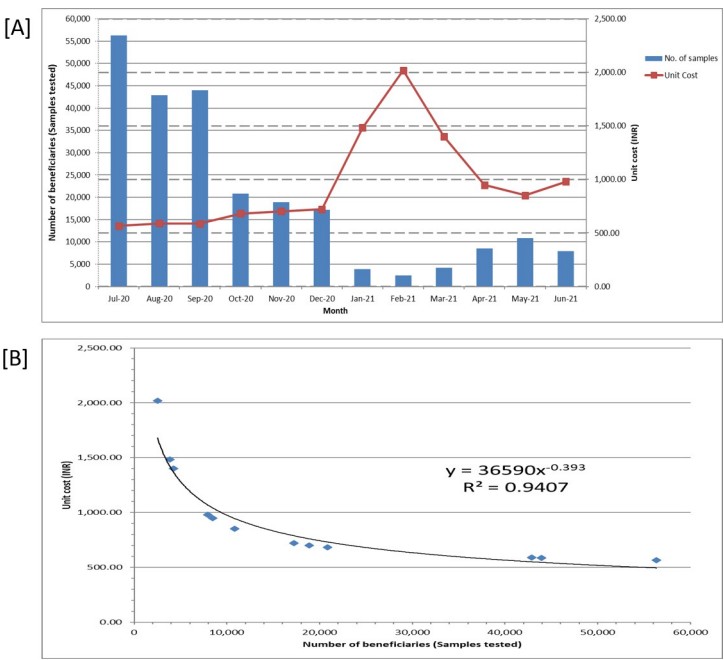

**Fig 3.** A] Correlation between number of samples tested and unit cost of COVID-19 rRT-PCR test, B] Regression analysis between number of samples tested and unit cost of COVID-19 rRT-PCR test.

correlation between the per-unit cost of the COVID-19 test and the number of samples tested. There was a negative correlation between the unit cost and number of samples tested for COVID-19 laboratory diagnosis, such that when the number of samples tested was decreased by 25%, the unit cost rose by 3.9%, and vice versa.

## Discussion

In the current study, the per unit cost of providing the COVID-19 rRT-PCR test during the early stages of pandemic mitigation was estimated to be ₹566 ($7.5). The total annual operating cost and per unit cost of the COVID-19 rRT-PCR test were estimated to be approximately ₹164.4 million ($2.2 million) and ₹691 ($9.2), respectively.

We noticed that purchasing lab consumables accounted for the majority of the overall cost (87%), which is comparable with the findings of Jacobsen et al. 2021, who reported that the COVID-19 rRT-PCR test kit and consumables price contribute for up to 70% of the total cost [22]. Other WHO factsheets that include the unit cost of other COVID-19 rRT-PCR kits likewise indicate that the bulk of the money was spent on rRT-PCR kits and consumables [23]. The majority of the money spent on lab consumables (89%) went into acquiring COVID-19 rRT-PCR kits and reagents, with the rest going toward plastic-ware products (S1A Fig). The rising costs of rRT-PCR test might be attributed to the global COVID-19 epidemic, procurement of consumables without tendering on an emergency basis, a supply shortfall, increased demand, and the prohibition on reusing specific lab wear and personal protective equipment (PPE kits). A research conducted during the 2009 H1N1 flu pandemics found that the cost of providing health services was heavily impacted by the protective measures used [24]. Another explanation might be that in the Indian setting, most of the COVID-19 rRT-PCR kit components were imported since there were no indigenous choices available at the time. Human resource costs account for around 10% of overall costs, with the bulk (96.8%) of funds going to NIC employee pay and just 3.2% to AMU staff salaries. The number of workers deployed for the COVID-19 rRT-PCR test was about equal by kind of employment (regular/contractual) (i.e. 50.5% of the HR deployed was on contract basis and 48.5% were on regular basis). Despite the fact that both job types had almost comparable numbers of HR, permanent HR received a higher percentage of salary (65.6%) than contract personnel (34.4%) (Table 4). This might be attributable to differences in their pay structures, although their activity-based time allocation was almost identical. Salary allocation for NIC employees was based on time allocation for a certain task. In contrast, according to statistics on staff time allocation, AMU employee pay was further lowered to half during analysis based on the personal observation. The observation was that the AMU personnel was engaged in COVID-19-related accounting operations, such as shipping COVID-19 rRT-PCR kits and reagents to regional and state labs around the nation, as part of their COVID-19 job.

Human Resource (HR) was the second highest (10%) contributor to unit cost of COVID-19 rRT-PCR test as per our study. One recent study on costing of Tuberculosis (TB) diagnostics from Tamil Nadu, South India also suggested HR as a major contributor to unit cost of TB tests [25]. Labour cost was the maximum contributor for unit cost of Medical diagnostic services of Hormone section of the central laboratory in Iran's East Azerbaijan Province using activity-based costing method [26]. This discordance may be due to the high market pricing for COVID-19 rRT-PCR kits during the pandemic. And although, human resources utilization in health care was the biggest component of a hospital's overall operating cost, according to Chatterjee et al 2013 [27]. Additionally, they noted that the difference in wage structures between commercial and public hospitals has a significant impact on HR costs, which is consistent with our results regarding the salary structure of contract and permanent employees.

According to certain research on the economics of Indian hospitals, human resources account for the bulk of a hospital's overall operating costs [27–30], a finding that contrasts with our analysis. This disparity in the proportion of cost of human resource may be explained by the study's design, the methodology used to analyze cost data. Additionally, these studies conducted cost analyses for a variety of healthcare services provided by public sector hospitals in India, but we are just accounting for one diagnostic test.

Additionally, we discovered that when this laboratory's workload (number of COVID-19 sample testing) was high, some technical personnel was sent from other labs within the same institution to do COVID-19 rRT-PCR responsibilities in addition to their usual scientific operations. The apportioning data used to calculate their cost contribution were chosen solely on the basis of their time commitment to the COVID-19 rRT-PCR test. The maximum duty hours per day included in this costing study were eight hours; nonetheless, staffs worked around the clock to complete COVID-19 duties. The additional duty hours provided by personnel were not included into this pricing estimate. Secondly, if human resources were dedicating additional duty hours to executing the COVID-19 rRT-PCR test, the number of samples examined may have been greater; hence, the unit cost could have been impacted by this aspect. Furthermore, multitasking staff participated in a variety of COVID-19 events. Their time contribution to COVID-19 rRT-PCR laboratory testing was evaluated using an apportionment method based on the number of activities they completed and the number of scientific activities conducted concurrently in the NIC.

Non-laboratory consumables, such as stationary and associated products, accounted for around 0.01% of the entire cost of delivering the service, which was minimal. This might be because of the restricted usage of paper work during the pandemic to contain the spread of the SARS-CoV-2 virus and compliance with COVID-19 guidelines. The majority of record keeping at NIC was done electronically; nevertheless, stationary was utilized primarily for filling out patient forms, equipment log books, record registers, and preserving official documents. Although the AMU's usage of stationary and other non-laboratory consumables was not measured in this research due to a lack of specific spending data.

Utility costs have been approximated for both cost centres (NIC & AMU). Utility costs (electricity, water, telephone, internet, bio-waste disposal, and laundry) were projected to be less than 2% of total costs using monthly utility bills / pay slips. Electricity bills were not supplied individually for each building, but rather for the institution as a whole. The power consumption for the COVID-19 rRT-PCR test was estimated by enumerating all electrically operated components and their daily use (in hours); hence, the monthly energy consumption (in KW) was computed with assistance from the institute's engineering support personnel. However, whereas the literature on this impact proposes allocating utilities and utility charges based on building space/floor area, we found that our method was more precise in terms of cost computation [29–33]. The corporation provides free Internet access as part of a memorandum of understanding (MoU) between the two groups.

Laundry services were not employed owing to the tight adherence to COVID-19 regulations prohibiting the reuse of laboratory protective equipment. Basically, cost data for products offered for free (internet service in this instance) or donated equipment should have been included in the cost analysis, but since the research site did not maintain track of donated items, we removed them from our calculations. The institute's bio-waste was disposed of by an outsourced agency and was charged as 'COVID—waste' or 'Non-COVID waste'. COVID waste prices were found to be much higher and almost twice those for non-COVID waste. COVID waste disposal accounted for about 17.66% of overall utility costs, whereas energy accounted for the majority (77.8%) (S1C Fig). For AMU, all utilities were allocated depending on the number of HR personnel deployed and their time commitment to COVID-19 tasks.

The overall capital costs associated with delivering COVID-19 rRT-PCR services were only 1.6% of total costs, with laboratory equipment accounting for 0.7%, other instruments accounting for 0.25%, and building space accounting for 0.58%, respectively (Table 3). Capital costs was annualized throughout the asset's useful life to get the yearly cost equivalent. The laboratory equipment that contributed the most to overall capital cost was determined to be imported goods and automated systems (S1D Fig). However, prompt diagnosis is critical in limiting disease progression and development during the COVID-19 pandemic, the institution management may consider manual techniques rather than automated systems as alternate solutions to decrease consumable costs; however, turnaround time will be impacted. Due to sample overload, all costly equipment was employed to its maximum potential in this investigation, indicating that automated systems seem to be cost-efficient when sample loads are very high. The consumables required for the automated systems were likewise quite expensive, accounting for about 57% of the entire lab consumables cost. This clearly indicates that the overall consumable cost will be almost halved if the manual process is used in the event of a lower sample load.

Certain fixed capital expenses, such as land and building space prices, are outside the organization's control; nonetheless, in this analysis, building infrastructure contributed a negligible 0.58% to overall cost. This may be due to use of activity-based apportioning values while computing infrastructure cost. Also, this demonstrates the most effective use of available space, despite the area's high rental value. NIC contributed the majority (96%) of the building space costs, while AMU contributed just 4%. The AMU's contribution to building space was less because it was apportioned based on the number of AMU staff deployed for COVID-19 rRT-PCR-related activity as a percentage of total AMU HR and their subsequent allocation based on time preference, whereas the NIC's space was maximally utilized for providing COVID-19 rRT-PCR service.

Varying lab consumable costs by 50% and 75%, result in the unit cost to vary between 390.6 ($5.2) - 992.27 ($13.2) and 240.2 ($3.2) - 1142.6 ($15.4) respectively. Costs of lab supplies were initially expensive, and they were utilized for costing analysis during the first wave of the COVID-19 epidemic; however, prices were reduced over time as a consequence of bulk purchasing at the central level and price negotiation with commercial suppliers. Rather, such costs were not included in the research; however, we were able to present the potential per unit cost of the test if consumable prices varied by 50% to 75%, as noted before. The price decrease may also be a result of the availability of several indigenous laboratory consumables via the government of India's flagship 'Make in India' initiative. We utilized a standardized bottom-up costing technique because it produces more accurate results than a top-down approach [30, 34–36]. To account for seasonal change in COVID-19 cases, we collected data for a whole year. The negative correlation between the unit cost and number of samples tested, indirectly reveals the relation between the unit cost of RT PCR test and optimal resource utilization. The policy maker could easily observes the optimal utilization of the resources by following this negative correlation.

Our research has several limitations, such as the fact that additional supported services such as security, transportation, and food services for COVID-19 employees may have been included in this analysis. Additionally, the cost component of shared premises (such as hallways) was omitted from this research.

## Conclusion

Our study is the first to explore the per-unit cost associated with providing the COVID-19 rRT-PCR tests service at the National Reference Laboratory in India in the early phases of pandemic mitigation, as seen from the perspective of the service provider. It explores how costs change with the fluctuations in demand and provides an understanding of the cost drivers for

rRT-PCR testing. Keeping in mind the varying levels of COVID-19 rRT-PCR service providing in India, further costing studies need to be carried out on a larger scale in order to acquire a clearer image of the government's spending and provide more complete information for policy objectives. Future economic evaluation studies on COVID-19 diagnostic techniques can be based on our estimates of per-unit costs of rRTPCR Test. The stakeholders may use these projections to plan for a comparable level of laboratory infrastructure. Our study would help the Government to understand the value for money they invested for laboratory diagnosis of COVID-19, budget allocation, integration and decentralization of laboratory services so as to help for achieving universal health coverage.

## Supporting information

**S1 Fig.** A] Cost distribution among laboratory consumables, B] Cost distribution among rRT-PCR reagents & kits, C] Cost distribution among overhead expenses, D] Cost distribution among laboratory equipments used.
(TIF)

**S1 Data. Data used to generate figures and graphs.**
(XLSX)

## Acknowledgments

We sincerely thank, Health Technology Assessment India (HTAIn), Department of Health Research, MoHFW, Government of India for necessary support. We also extend our gratitude to all COVID-19 staff at ICMR-National Institute of Virology (NIV), Pune for their help and cooperation during the data collection process. We acknowledge the financial support provided by Department of Health Research (HTAIn) in terms of human resource to Health Technology.

Assessment Resource Center at ICMR-NIV Pune, India.

## Author Contributions

**Conceptualization:** Naveen Minhas, Yogesh K. Gurav, Susmit Sambhare.

**Data curation:** Naveen Minhas, Yogesh K. Gurav, Susmit Sambhare.

**Formal analysis:** Naveen Minhas, Yogesh K. Gurav, Susmit Sambhare.

**Investigation:** Naveen Minhas, Yogesh K. Gurav, Manohar Lal Choudhary, Sumit Dutt Bhardwaj.

**Methodology:** Naveen Minhas, Yogesh K. Gurav, Susmit Sambhare, Varsha Potdar.

**Project administration:** Yogesh K. Gurav, Priya Abraham.

**Resources:** Yogesh K. Gurav, Manohar Lal Choudhary, Sumit Dutt Bhardwaj.

**Software:** Yogesh K. Gurav.

**Supervision:** Yogesh K. Gurav, Susmit Sambhare, Varsha Potdar, Manohar Lal Choudhary, Priya Abraham.

**Writing – original draft:** Naveen Minhas, Yogesh K. Gurav, Susmit Sambhare.

**Writing – review & editing:** Naveen Minhas, Yogesh K. Gurav, Susmit Sambhare, Varsha Potdar, Manohar Lal Choudhary, Sumit Dutt Bhardwaj, Priya Abraham.

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
