## [Decision Letter · Decision Letter 0]

4 Aug 2022

PONE-D-21-40768Cost-Analysis of Real Time RT-PCR Test Performed for COVID-19 Diagnosis at India's National Reference Laboratory During the Early Stages of Pandemic Mitigation.PLOS ONE

Dear Dr. Gurav,

Thank you for submitting your manuscript to PLOS ONE. After careful consideration, we feel that it has merit but does not fully meet PLOS ONE’s publication criteria as it currently stands. Therefore, we invite you to submit a revised version of the manuscript that addresses the points raised during the review process.

ACADEMIC EDITOR: Please revise the paper to incorporate the comments and suggestions of the reviewers. The language used in the paper may be considerably improved; however, need for a native English expert may not be warranted at this time as this may be done by authors themselves. 

We look forward to receiving your revised manuscript.

Kind regards,

Aneesh Basheer

Academic Editor

PLOS ONE

https://journals.plos.org/plosone/s/file?id=ba62/PLOSOne_formatting_sample_title_authors_affiliations.pdf".

2. Please provide additional details regarding participant consent. In the ethics statement in the Methods and online submission information, please ensure that you have specified (1) whether consent was informed and (2) what type you obtained (for instance, written or verbal, and if verbal, how it was documented and witnessed). If the need for consent was waived by the ethics committee, please include this information.

3. PLOS requires an ORCID iD for the corresponding author in Editorial Manager on papers submitted after December 6th, 2016. Please ensure that you have an ORCID iD and that it is validated in Editorial Manager. To do this, go to ‘Update my Information’ (in the upper left-hand corner of the main menu), and click on the Fetch/Validate link next to the ORCID field. This will take you to the ORCID site and allow you to create a new iD or authenticate a pre-existing iD in Editorial Manager. Please see the following video for instructions on linking an ORCID iD to your Editorial Manager account: https://www.youtube.com/watch?v=_xcclfuvtxQ.

Additional Editor Comments:

Please revise the paper according to the comments of the reviewers. English may be improved considerably although this may not require the use of a native English expert as suggested.

Reviewers' comments:

Reviewer's Responses to Questions

**Comments to the Author**

1. Is the manuscript technically sound, and do the data support the conclusions?

Reviewer #1: Yes

Reviewer #2: Yes

2. Has the statistical analysis been performed appropriately and rigorously? 

Reviewer #1: Yes

Reviewer #2: Yes

3. Have the authors made all data underlying the findings in their manuscript fully available?

Reviewer #1: No

Reviewer #2: Yes

4. Is the manuscript presented in an intelligible fashion and written in standard English?

Reviewer #1: No

Reviewer #2: Yes

5. Review Comments to the Author

Reviewer #1: Cost-Analysis of Real Time RT-PCR Test Performed for COVID-19 Diagnosis at India's National Reference Laboratory During the Early Stages of Pandemic Mitigation

Many thanks for the opportunity to read and review an important costing study for India’s national reference laboratory network. A lot of work has gone into this study and it generates some important findings for planning and prioritising COVID testing strategies in India – one of the country’s worst hit by the COVID pandemic.

My recommendation is to make revisions in the write up of the paper including having an English language speaker/editor review the paper to ensure that the language is correct. The methods write up requires particularly attention.

My comments are as follows:

Authors and disclosures section:

- Financial disclosure: you state that the authors received no specific funding for this work; however if this was done in the course of/as part of their responsibilities work at their respective institutions then this should be acknowledged (i.e. their employers funded the work). I note that you acknowledge the support of HTAIn and ICMR in the acknowledgements section.

- Data availability: please state where the data is available. Most journals accept that the data is available with the author on request but you do need to state this. Please double check you are following the publication guidelines on this.

- Authors: normal practise is to state what role each author took in the analysis (writing, analysis, critical review and contribution to content, design, data collection etc etc).

Abstract:

- please make sure that the time frame is specified for the unit cost and for the number of tests.

Background

- please clarify your objectives in this section so that the reader knows what you are doing – a lot of your analysis comes as a surprise when you read through the paper. Not only are you looking at the unit cost (as you state) but you are doing so for a peak period in the pandemic; and then you are estimating the average annual cost for PCR testing as well as an estimated average annual unit cost for testing that takes account of peaks and troughs in the pandemic. You also use your data to understand the breakdown of costs across inputs and over time and different levels of testing activity. Please state all these objectives upfront.

- The change in unit cost as the pandemic progresses and the correlation between unit cost and outputs is a really important aspect of the study. Please introduce this issue and explain why this is important in the background section.

- Pg1 line 57 – why “like” this test? Isn’t it this test that is gold standard?

- Lines56-60 – you state twice the test is gold standard.

- Pg 1 line 64 – please specify that you mean the Indian government

- Lines 64-66 - This is not stated in the article referenced. Please access better data on the spend on testing relative to the health budget or explain the important impact that testing has on overall health care expenditure in a different way. E.g. The government has spent xxxxUSD on the establishing COVID-19 rRT-PCR labs, recruiting……

- Pg2 lines68-70 – you need to help the reader more here. Please explain what you mean by economic and financial evaluations and efficient resource allocation? (This is not a health economics journal)

- Pg2 lines 72-73 – can you provide some examples of how this has been stated. Also these costs are likely based on prices charged. You need to describe the problem with using charges as an estimate of costs - i.e. they are related to the business model of the provider/lab and have profit margins built in and don't necessarily reflect the actual cost of production. And why do we need to know the cost of production.

- Lines 75 – the Gold reference seems out of place – it tells us what gross costing is but doesn’t tell us what method has been used in this instance.

- Line 79. Start a new sentence after country[12].

- Lines79-82 I don't follow this logic. Are the new labs going to replace state labs? What is the relevance for your costing study. Your rationale is better focussed on the lack of data for planning the new laboratories and ensuring adequate investment as well as for use in economic evaluations of different strategies for COVID-19 testing.

Methodology: Study design and study site

- Please add a description of the intervention; is this lab only costs or is sample taking also included? What are the procedures followed when testing? How long does it take to return a result? How do the different cost centres relate to the testing? Are these dedicated COVID testing sites - what other testing is carried out? How many tests are carried out per month/year? Please also describe the roles of the different cost centres.

- In addition please state and explain the approach to costing here (activity based costing, bottom up etc)

Methodology: Data collection

- Line 100 delete “form the competent authorities”

- Line 103 cost centers – should be cost center (there is only one other)

- Line 105, line 109, line 112 – please avoid using etc – be specific

- Line 111 delete “in the proforma… …laboratory” and just state “The data was collected from the stock registers, instrument……

- Line 113 – What is comprehensive information? What information was collected - i.e. numbers of staff, categories of staff and when they were present?

- Line 118-9 – please clarify if staff time include both admin and procedure time for each individual staff member?

- Table 1 - Are all of these costs specific to rRT-PCR testing or are some (e.g. HR) used for other activities as well? If the latter - it would be important to have a column which explains how you allocate these resources to the testing. e.g. how do you know how many of each lab consumable were specifically used for PCR testing?

- The allocation methods are described partially in different places in the methods section, in table 2, in the results section, and the discussion section. Please bring these all together into one place so that it is clear how these allocations were made and what data sources were used.

- Please add a section within the Methodology called “Allocation of resources” or similar; you could then have a separate section on “Price/value data” in which you describe all the price data sources.

- Line 153-4 – move the reference to activity based costing and what this is to the study design section

- Line 164-165 please make sure it is clear whether this allocation is at the individual or overall level

- Line 188 is where you start the description of the analysis - Add a section here called “Analysis”

- Please add to your “analysis” section a description of how you: calculate early pandemic unit cost; unit cost for each month; total average annual operating cost that accounts for fixed and variable cost; analysis of relationship between service output and unit cost

- Also add that you do a detailed descriptive analysis of the breakdown of costs by inputs.

- Please be consistent in your terminology – sometimes you refer to annual expenditure, sometimes annual operating cost sometimes total cost. Avoid using the word expenditure when doing a cost analysis as this implies you are looking at total value of financial transactions which generally a costing is not trying to do.

- Explain why the fixed and variable cost classification is required i.e. to help estimate the cost variation by month and the average annual cost as they behave differently with respect to different output levels and the nature of the pandemic has led to fluctuations in output.

Methodology: Table 2

- different types of HR are allocated differently (according to the text) and this is not evident in the table; I suggest you separate the HR into rows categorised according to how the staff time was allocated

- It is not clear how reference 13 could be used for allocating staff time? Why was this used and not the observational work? Or interviews with staff at the laboratories.

- I suggest you have a column which states the source of the data (e.g. interview, records etc)

- Please explain why you need an assumption and then what the assumption is e.g. for HR at AMU and also explain why this is a reasonable assumption

- Does indent mean item? Please use an alternative word to indent for an international audience. And please state the source for the consumption data (is this reference 13? If it is reference 13 then please explain in a footnote how this relates to your study.

- Lab equipment – why was time used here and not number of tests relative to all tests? Did you account for time outside normal working hours?

- Rows 6-9 column 2– so is this based on the information from rows 1 and 2 (i.e. HR)

Sensitivity analysis

- Please state which result you are testing in the sensitivity analysis methods section; is it the unit cost or total cost?

- Why did you choose these particular variables?

- Could you also look at the impact of variations in staff time (or at least the most important staff members) on the overall cost.

- Can you explain the choice of the 25% range used; and do you apply this to the overall figure or at the level of the individual input?

- You refer to some of the justifications around the choice of variables for the sensitivity analysis in the discussion – this would be better placed here in the methods section.

Results

- Line 216 – is annual operating cost what you refer to as expenditure in the methods?

- Table 3 please add the timeframe to the title and indicate which costs are reported in the table.

- Table 3 - For fund distribution column – is this percentage of unit cost or total cost? I am assuming that the unit cost is the peak month cost and the total cost is the extrapolated cost and therefore the % breakdown would be different. But this is also not clear and needs to be labelled in the table.

- Table 3 - Tidy up table. No need for decimal places; use comma after each three digits, total cost might be better presented in thousands. Align numbers to right of column for numbers; left of column for text.

- Table 3 - For the cost breakdown, could you add sub rows for the important cost elements to this table and then eliminate the need for the figure 1 (or figure 1 can go in appendix)

- The recurrent cost analysis is not referred to in the methods section -please so so.

- Please extract all references to methodology from the results and ensure that these are documented in the methods section and avoid duplication e.g lines 230-231; 258-9; 269-70

- Table 4 - what is the cost contribution of the different types of staff? can you add this? It would be an interesting addition; are the costs here monthly or annual (and/or relating to peak pandemic)

- Figure 2 could be tidier….. how about using stacked bar charts? Or use a legend rather than all the label and lines?

- Figure 3 – the axes are a bit off; you should be able to cut them at a point >0 so you don’t have such a big gap. The legend needs more detail.

- Line 293 -295 – a percentage figure is reported but its not clear what this represents – does this relate to the percentage difference resulting from the sensitivity analysis?

- Line 296 – What do you mean by does not change much? Is this relative to something?

- Line 298-99 – why is this regression not mentioned before?

- Line 300-301 – please explain either in the methods or in the discussion why this negative correlation is so important particularly in a situation with fluctuations in demand.

Discussion

Line 317 – Which rising costs? Its not clear what you are referring to here.

Line 329 - how have you calculated salary? did you include benefits? which is important for comparison with contract personnel.

Line 332-3 – in what context was the pay lowered to half – in practice or for the analysis? And why?

Line 336 – you need to justify why overall hospital costs and their structure are relevant here when the costing topic here is related to diagnostics and laboratories which will necessarily be very different (e.g. there is no patient care). The only point of relevance is that on wage structures.

Line 340-343 – hospital cost structures are not relevant to your analysis; they provide completely different services so please remove this sentence. Please find some diagnostics costings and then make the comparisons.

Line 352-3 – its an important point that you did not include the additional hours of staff during the pandemic. Please can you add a scenario to your sensitivity analysis that accounts for this. I believe your unit cost will rise for the peak pandemic periods and bring it more in line with the months of low testing rates.

Pg 17-21 – there is one long paragraph – this is too long. Please split this into separate paragraphs that make separate points. Also don't repeat methodology – this should all be in the methods section. There's too much detail here that should either be in the methodology or in a supplement. When you’re identifying the key limitations, state why its a problem and what you've done to mitigate this e.g. sensitivity analysis and whether its an important bias or not. If there are multiple limitations you may need more than 1 paragraph.

Pg line 397-8 One what basis is this cost-effective? How does being used to its maximum potential imply cost-effectiveness?

Line 405 – The alternative possibility for what? Please explain.

Line 407 – without the context of what the AMU function is, its difficult to understand whether this is important or not? Please explain under the study setting section.

Line 409 (and elsewhere) – write percentage out in full

Line 411-413; line 422-23 – this is all methods and belongs in the methodology

Line 426-7 – so do you mean that these items have been excluded? What are the implications of this?

line 429 -431 – the study does more than this – it also explores how costs change with the fluctuations in demand and provides an understanding of the cost drivers for PRC testing.

Overall the discussion is very much focussed on the limitations and methodological weaknesses – can you add something on the policy implications of the findings – particularly from figure 4 and the HR issues (e.g. is there excess capacity to ensure the testing is carried out or are staff working over time and not been recompensed) .

Typos:

Pg1 Line 50 – add “the” after pandemic.

Pg1 line 54 – what do you mean by suspect?

Pg1 line62 – delete “being”

Pg2 line 78 – add “The” before Government

Reviewer #2: The manuscript is a very welll researched cost-analysis of Real Time RT-PCR Test performed for COVID-19 diagnosis in the National Reference laboratory. The authors have detailed every componenent of the work process, including personnel, infrastructure, equipment and running cost .

Points to note:

1. The cost of training the personnel who were both contractual and full time from other sections,and were used to perform highly technically exacting molecular tests, has not been analysed.

2 The following statement needs to be justified for setting up similar laboratories throughout the whole country as suggested by the authors "Our study estimates can be used to ascertain the cost effectiveness of free of cost provisioning of this service by the government and can also be used for setting up a similar level of lab infrastructure throughout the country".

The Covid pandemic saw all Govt. tertiary care and headquarter hospitals set up molecular laboratories to carry out Real Time RT PCR tests and were performing large number of tests with existing manpower. So workable laboratories have already been set up.

3. The Central Govt. Health and Research Department (ICMR) has already set up several Virus research Diagnostic laboratories across the country. The authors could advocate cost analysis keeping in mind the already existing VRDLs.

Do the authors advocate more such laboratories to the level of the National Reference Lab ?

6. PLOS authors have the option to publish the peer review history of their article (what does this mean?). If published, this will include your full peer review and any attached files.

Reviewer #1: No

Reviewer #2: **Yes: **Reba Kanungo

---

## [Author Response · Author response to Decision Letter 0]

4 Oct 2022

Rebuttal Letter [PONE-D-21-40768]

We thank the academic editor and the two reviewers for their valuable comments on our manuscript. Our response to each point raised by the academic editor and reviewers is mentioned below. We hope that we have addressed all the points up to the satisfactory level and the manuscript will now be suited for the publication.

Sincerely,

On behalf of all authors,

Dr. Yogesh Gurav

---

## [Editor Report · Decision Letter 1]

7 Nov 2022

Cost-Analysis of Real Time RT-PCR Test Performed for COVID-19 Diagnosis at India's National Reference Laboratory During the Early Stages of Pandemic Mitigation.

PONE-D-21-40768R1

Dear Dr. Gurav,

We’re pleased to inform you that your manuscript has been judged scientifically suitable for publication and will be formally accepted for publication once it meets all outstanding technical requirements.

Kind regards,

Aneesh Basheer

Academic Editor

PLOS ONE

Additional Editor Comments (optional):

Thank you for revising the paper incorporating all comments and addressing all concerns of the reviewers.
---

## [Editor Report · Acceptance letter]

10 Nov 2022

PONE-D-21-40768R1 

Cost-analysis of real time RT-PCR test performed for COVID-19 diagnosis at India's National Reference Laboratory during the early stages of pandemic mitigation. 

Dear Dr. Gurav:

I'm pleased to inform you that your manuscript has been deemed suitable for publication in PLOS ONE. Congratulations! Your manuscript is now with our production department. 

Kind regards, 

on behalf of

Dr. Aneesh Basheer 

Academic Editor

PLOS ONE